# A POI and LST Adjusted NTL Urban Index for Urban Built-Up Area Extraction

**DOI:** 10.3390/s20102918

**Published:** 2020-05-21

**Authors:** Fei Li, Qingwu Yan, Zhengfu Bian, Baoli Liu, Zhenhua Wu

**Affiliations:** 1School of Environment Science and Spatial Informatics, China University of Mining and Technology, Xuzhou 221116, China; Lifei@cumt.edu.cn (F.L.); Zfbian@cumt.edu.cn (Z.B.); liubaoli@cumt.edu.cn (B.L.); wuzhenhua@cumt.edu.cn (Z.W.); 2Jiangsu Key Laboratory of Resources and Environmental Information Engineering, China University of Mining and Technology, Xuzhou 221116, China

**Keywords:** urban built-up area, nighttime light remote sensing, NPP/VIIRS, Luojia 1-01, POI, LST

## Abstract

Nighttime light (NTL) images have been broadly applied to extract urban built-up areas in recent years. However, the typical NTL images provided by Defense Meteorological Satellite Program/Operational Linescan System (DMSP/OLS) and National Polar-Orbiting Partnership’s Visible Infrared Imaging Radiometer Suite (NPP/VIIRS) have the drawbacks of low resolution and blooming effect, which bring difficulty for the application of them in urban built-up area extraction. Therefore, this paper proposes the POI (point of interest) and LST (land surface temperature) adjusted NTL urban index (PLANUI) to extract the urban built-up areas with high accuracy. PLANUI is the first urban index to integrate POI and NTL for urban built-up area extraction. In this paper, NPP/VIIRS and Luojia 1-01 images were introduced as the original NTL data and the vegetation adjusted NTL urban index (VANUI) was selected as the comparison item. The threshold method was utilized to extract urban built-up areas from these data. The results show that: (1) Based on the comparison with the reference data, the PLANUI can make up the shortcoming of low resolution and the blooming effect of NTL effectively. (2) Compared with the VANUI, the PLANUI can significantly improve the accuracy of the urban built-up areas extracted and characterize urban features. (3) According to the results based on NPP/VIIRS and Luojia 1-01 images, the PLANUI has extensive applicability, both for regions with different degrees of economic development and NTL data with different resolutions. PLANUI can enhance the features of urban built-up areas with social sensing data and natural remote sensing data, which helps to weaken the NTL blooming effect and improve the extraction accuracy. PLANUI can provide an effective approach for urban built-up area extraction, which plays a certain guiding role for the study of urban structure, urban expansion, and urban planning and governance.

## 1. Introduction

The nighttime light (NTL) data can capture light signals from urban buildings, road facilities, and vehicles. The NTL images in urban built-up areas have continuous spatial distribution and brightness value significantly higher than that in surrounding areas. Therefore, many studies utilize it for the urban built-up area extraction [1,2,3,4,5,6]. Currently, the typical NTL images are provided by Defense Meteorological Satellite Program/Operational Linescan System (DMSP/OLS) and National Polar-Orbiting Partnership’s Visible Infrared Imaging Radiometer Suite (NPP/VIIRS) [7,8]. Their spatial resolutions are 1 km and 500 m, respectively. However, the study of extracting urban built-up areas using NTL data only is limited to a large scale due to the low resolution, which makes it hard to obtain high-precision urban built-up area results on small and medium levels [9]. Meanwhile, the area of the urban built-up area extracted from the NTL image is larger than the actual range due to the blooming effect [10], and the extraction accuracy is low. Elvidge et al. [2] also believe that the blooming effect is one of the main causes of overestimating the urban built-up areas.

In recent years, many studies [11,12] combined multi-source data with NTL data to build urban indices for the urban built-up area extraction to improve its accuracy on small and medium levels. Several studies have demonstrated that abundant information on urban built-up areas can be obtained by utilizing multi-source data with different characteristics [13,14]. The human settlement index (HIS), vegetation adjusted NTL urban index (VANUI), and enhanced vegetation index (EVI) adjusted NTL index (EANTLI) are the most broadly utilized urban indices. They assume an inverse relationship between vegetation and urban built-up area to mitigate NTL blooming effect and characterize urban built-up areas [15,16,17]. Lu et al. [15] proposed the HIS by combining NTL images with the normalized difference vegetation index (NDVI) images. HIS overcorrects the light signals in peri-urban areas. Besides, NTL blooming effect is still obvious in bare soil areas, where NDVI values are zero. Zhang et al. [16] proposed VANUI combining NDVI and NTL, which enriches urban fringe information and is broadly utilized in urban built-up area extraction. However, VANUI has a limitation in the peri-urban areas where both vegetation values and NTL values are high. In addition, it is not suitable for some desert cities in North America regarding the unobvious relationship, between urban built-up areas and vegetation. According to the same principle, Zhuo et al. [17] built the EANTLI by combining EVI with NTL. Compared with NDVI, EVI can weaken the effects of atmosphere and soil background on vegetation index. Therefore, it can promote the accuracy of the urban built-up areas extracted without the shortcomings of NDVI. However, the EANTLI value may be abnormally high especially for the mixed pixels in the water land boundary region, which increases the misclassification error.

Previous articles ignored combining NTL and point of interest (POI) data to build an index for urban built-up area extraction, while actually POI is positively correlated with urban built-up areas. POI is a kind of social sensing data produced by human activities and contains a wealth of location and attribute information. The abrupt changes of its density at the boundaries between urban and surrounding suburbs and rural regions make it easier to extract urban built-up areas [18,19]. At present, the main method using POI is to set a threshold value for the kernel density of POI to obtain accurate urban built-up area results. Some studies have demonstrated that there is a good coupling relationship between NTL and POI, which has high consistency and broad applicability in the study of urban spatial structure [20,21]. POI brings convenience in obtaining the boundary of urban built-up areas accurately and helps to make up the shortages of the low resolution and the blooming effect of NTL data. Therefore, this paper combined NTL and POI data to build a new index for urban built-up area extraction. It is urgent to clarify the effects of the combination on the extraction improvement. In addition to POI, this paper also introduces a natural remote sensing data, land surface temperature (LST). Many studies have shown that the land surface temperature is positively correlated with the distribution of urban land cover [22,23]. The characteristics of urban built-up areas can be enhanced by LST from a natural perspective, which is different from the humanistic and social perspective of POI. Some [14,24,25] chose the combination of NTL and LST to extract urban built-up areas. Among them, He et al. [24] combined the NTL with LST to extract dynamic information of global urban expansion by the fully convolutional network. Further, Zhang et al. [25] proposed the temperature and vegetation adjusted NTL urban index (TVANUI) for the purposes of characterizing urban built-up areas and reducing the blooming effect. Their satisfactory results proved that the combination of LST and NTL has great potential in improving the extraction accuracy of urban built-up areas. Therefore, this paper combined NTL with POI and LST, to establish the POI and LST Adjusted NTL Urban Index (PLANUI) for the study on the extraction of urban built-up areas.

The widely used NPP/VIIRS images with 500-m resolution were introduced into this study to explore the effects of PLANUI. Moreover, the Luojia 1-01 satellite was successfully launched in June 2018 and began to provide NTL images with 130 m resolution and 250 km width. This data greatly enhances the spatial resolution of NTL and was proved to have a great ability to extract urban areas [26,27,28]. Therefore, this paper also introduced Luojia 1-01 images into the urban built-up area extraction experiments to verify whether PLANUI is suitable for NTL data with increasingly high spatial resolution.

The primary purpose of this paper is to propose the PLANUI that combines NTL images with POI and LST data to reduce the blooming effect of NTL and enhance urban built-up areas features for promoting the extraction accuracy. The secondary objective is to explore whether PLANUI is suitable for NTL with a high resolution like Luojia 1-01.

## 2. Data

### 2.1. Study Area

The study selected Nanjing, the provincial capital of Jiangsu province of China, as the study area (Figure 1). Among the 11 districts under the jurisdiction of Nanjing, the main urban area, such as Gulou and Xuanwu, have relatively advanced economic development, while other administrative regions such as Lishui and Gaochun have remote geographical locations and relatively weak economic development. Through utilizing the characteristics of the high spatial heterogeneity of Nanjing’s regional development level, the broad applicability of PLANUI in regions with different development levels can be verified.

### 2.2. Data Preparation

NTL data includes Luojia 1-01 and NPP/VIIRS images (Figure 2a,b), the selection date of which is July 2018. Luojia 1-01 is provided by the High-Resolution Earth Observation System of the Hubei Data and Application Center. NPP/VIIRS is provided by the National Geophysical Data Center (NGDC). Table 1 shows the specific parameters of Luojia 1-01 and NPP/VIIRS.POI was crawled in May 2018 through the Amap API. After data cleaning, Nanjing’s POI totaled 366,123, divided into 13 categories, mainly including catering, shopping, culture, and life. Kernel density estimation was used to pre-process the POI data.LST (resolution 1KM, Figure 2c) is derived from the MODIS eight-day composite product (MOD11A2) in July 2018 provided by NASA (http://ladsweb.nascom.nasa.gov), the accuracy of which is better than 1 °C [29].The NDVI comes from the MOD13Q1 product (http://modis.gsfc.nasa.gov) provided by NASA. It has a 16-day temporal resolution and a 250 m spatial resolution. Data in June–September 2018 with the best effect was selected for averaging and min-max normalization.The reference built-up areas data is provided by the Resource and Environment Data Cloud Platform (http://www.resdc.cn/). It has a 100 m spatial resolution and is produced by visual interpretation and field investigation.

## 3. Methods

### 3.1. PLANUI: The POI and LST Adjusted NTL Urban Index

Previous studies [18,19,20,21,22,23,24,25,26,30] proved that POI data and LST data are suitable for the extraction of urban built-up areas and that combining the two with NTL data for the study of the extraction of urban built-up areas is feasible. Therefore, this paper combined NTL, POI, and LST data to propose PLANUI. After referring to related researches [14,25,31], the three data are given equal weight. PLANUI aims to make up the shortcomings of low spatial resolution and blooming effect of NTL images with the help of POI data and LST data, to obtain more accurate urban built-up area results. Based on the fact that urban built-up areas have generally higher LST values and POI density than other land types, PLANUI can enhance the signals of urban built-up areas and weaken the interference light signals of non-built-up areas, to improve the accuracy of the urban built-up areas extracted. Since the NTL DN (digital number) values, the POI density values, and the LST values have significant positive correlations with urban built-up areas, the “average value method” was chosen to establish PLANUI to utilize the positive correlation for the ideal results. The geometric average value in the “average value method” was selected for the following considerations: Firstly, the magnitude difference between the DN value of NTL, the kernel density value of POI, and the value of LST is tremendous. The geometric mean value can eliminate the impact of this difference and integrate the advantages of the three. Secondly, there are many noise points in the NTL images, which have higher DN values than actual. There are few or no points of interest near the noise points, so the POI kernel density value of the points approach zero. At the same time, the LST values at the points are also usually low. The geometric mean can use the POI kernel density values and LST values of the noise points to reduce the influence of the abnormal DN values on the calculation result of PLANUI. Thereby, it can avoid some areas with noise points being mistakenly extracted as urban built-up areas. According to the same principle, it can reduce the part of the results that are incorrectly extracted, such as roads and railways outside urban built areas. Thirdly, the urban built-up areas with weak light signals have a serious problem of missing information in the process of extraction. The DN values of NTL are low in areas with low economic development. In the process of extracting urban built-up areas using NTL images, these areas are easily overlooked, resulting in many missing parts in the extraction results. However, these regions have high POI and LST values as other regions. Therefore, the POI and LST values can be used to enhance the signals of urban built-up areas in the areas, and then make up the missing built-up areas. Fourthly, one of the characteristics of the geometric mean is that it is less affected by extreme values, which can correct the abnormal light values.
(1)$${\mathrm{PLANUI}}_{i}=\sqrt[{3}]{{\mathrm{NTL}}_{i}{\times P}_{i}{\times T}_{i}},$$
where ${\mathrm{NTL}}_{i}$ represents the DN value of point i, $P_{i}$ represents the POI kernel density value of point i, and $T_{i}$ is LST value at point i. In this paper, the PLANUI includes the LJ-PLANUI and the NPP-PLANUI according to the combined NTL data (Figure 3).

### 3.2. The Implement of PLANUI

The original NTL and LST data were preprocessed by necessary steps to avoid the abnormal pixels influencing the final results. This study used the method of precise geometric correction proposed by Jiang et al. [28], which utilizes distributed ground control points (GCPs) and Landsat 8 OLI to conduct the ortho-rectification, to correct the Luojia 1-01 images. In addition, the outliers of the LST data were removed by the quality control flags [25].

Compared with POI data, NTL and LST data are easy to preprocess before constructing PLANUI. It is challenging to obtain rational data of POI density. Therefore, kernel density estimation was selected to get the POI probability density in this paper. The advantage of this method is that it is not affected by grid size and position [32], so it can obtain high-quality POI kernel density estimation data. Kernel density estimation takes a regular area with a specific bandwidth near any point as the range of the density calculation and analyzes the spatial distribution status of the research object through the calculation results. The weight is given according to the distance from the center point. The weight of the data points close to the center point in the calculation area is relatively high. Otherwise it is low. So, the results obtained of all points in the study area are the weighted average density values [33]. The formula for calculating the kernel density $P_{i}$ is as follow:(2)$$P_{i}=\frac{1}{{n\pi R}^{2}}\times \overset{n}{\underset{j=1}{\sum }}K_{j}{(1-\frac{D_{\mathrm{ij}}^{2}}{R^{2}})}^{2}$$
where $K_{j}$ represents the weight of point j, $D_{\mathrm{ij}}$ represents the Euclidean distance between point i and point j, R represents the bandwidth ($D_{\mathrm{ij}}$< R) of regular area, and n represents the quantity of point j in the regular area.

The rational choice of bandwidth R according to the research question has a crucial impact on the results [34]. Because this paper needs to combine the POI kernel density data with two night-time light data with different resolutions, an appropriate bandwidth R should be selected. According to the current research results [31], a bandwidth of 1000 m is selected, and at the same time, one-tenth of the bandwidth is used as the side length of the resulting grid unit. The result is shown in Figure 2d.

### 3.3. The Evaluations of PLANUI

To fully evaluate the proposed index PLANUI, the widely used index VANUI was introduced as a comparison term. VANUI is a standardized urban index adjusted by vegetation proposed by Zhang et al. [16]. On the basis that there is a negative correlation between vegetation and urban surface, MODIS NDVI is used to weaken the NTL blooming effect and strengthen the characteristics of urban cores at night. The VANUI value of the core urban area approaches 1, and the VANUI value of non-urban and non-illuminated areas approaches 0, so the VANUI is widely used in the extraction of urban built-up areas.
(3)VANUI=(1−NDVI)×NTL
in this paper, the VANUI index includes the LJ-VANUI index and the NPP-VANUI index, according to the combined NTL data.

#### 3.3.1. The Extraction Method

The threshold method [35,36] is typical in the field of urban built-up area extraction based on NTL. This paper adopted the statistical data comparison method [37] with simple operation and high accuracy. This method uses statistical data to assist in selecting the threshold and uses the dichotomy to continuously change the threshold to make the extraction results approach the statistical data. The extraction result based on the optimal threshold is the urban built-up area. This method can make up the drawback of the blooming effect to a certain extent [38,39].

#### 3.3.2. Accuracy Assessment

The accuracy of the extraction results was quantitatively evaluated by the statistical classification index precision (Equation (4)), recall (Equation (5)), F1 score (Equation (6)) (the F1 score is the harmonic mean of precision and recall) [40], and Kappa coefficient.
(4)$$\mathrm{precision}=\frac{T_{P}}{T_{P}{+F}_{P}}$$
(5)$$\mathrm{recall}=\frac{T_{P}}{T_{P}{+F}_{N}}$$
(6)$$F1-\mathrm{score}=2\times \frac{\mathrm{prcision}\times \mathrm{recall}}{\mathrm{precision}+\mathrm{recall}}$$
where $T_{P}$ is the area of the correct part of the extraction results, $F_{P}$ is the area of the wrong part of the extraction results, and $F_{N}$ is the area of the omission pixels.

The workflow of conception, implementation, and evaluation of PLANUI is shown in Figure 4.

## 4. Results

### 4.1. Comparison of Spatial Distributions

Figure 5 shows the spatial distribution of Luojia1-01, LJ-VANUI, and LJ-PLANUI in all administrative regions of Nanjing. The black line represents the reference urban built-up area and is superimposed on these images. With the advantage of high spatial resolution, Luojia 1-01 images can reflect many details, such as streets and roads. It also enables luojia1-01 to roughly reflect the general distribution of urban built-up areas. However, within the urban built-up areas, the DN value of it varies greatly, which fails to accurately reflect the continuous distribution characteristics of urban built-up areas. Meanwhile, the blooming effect exists in some areas with high light value (Figure 5a,b). There are also many interference light signals from roads and non-built-up areas in Luojia 1-01 image. LJ-VANUI slightly enhanced the signals of surrounding administrative areas and weakened the signals of surrounding roads. In general, the changes brought by LJ-VANUI are not noticeable. LJ-PLANUI reduced the gap of DN value by enhancing the missing light signals in urban built-up areas. Meanwhile, the light signals in the non-built-up areas are weakened to reduce the influence of blooming effect and interference lights.

NPP/VIIRS has the blooming effect, interference lights, and weak light signals in local areas (Figure 5a–c). In the Lishui district and Gaochun district, the problem of light signals missing is serious (Figure 5d,e), which is caused by the low spatial resolution of NPP/VIIRS. NPP-VANUI mainly corrects the concentrated and high light signals in the main urban area. The improvement brought by NPP-VANUI is not obvious. NPP-PLANUI reduces the blooming effect and interference lights by weakening the light signals outside urban built-up areas. Meanwhile, NPP-PLANUI increased the missing light signals of urban built-up areas, especially in the Lishui district and Gaochun district. In a word, PLANUI effectively corrects the blooming effect and reduces the interference information while increasing the missing information.

### 4.2. Extraction Results

#### 4.2.1. Comparison between Extraction Results and the Reference

The final results of Nanjing were extracted from Luojia 1-01, NPP/VIIRS, LJ-VANUI, NPP-VANUI, LJ-PLANUI, and NPP-PLANUI by thresholds 17,450, 16.2, 9200, 8.65, 2200, and 230, respectively. These results were compared with the reference, as shown in Figure 6.

The overall shape of the extraction results from the Luojia 1-01 image is basically consistent with the reference data. However, the urban built-up areas are fragmented and have many holes. Much other non-built-up area information, such as roads outside urban built-up areas, is also extracted by mistake. The results extracted from the LJ-VANUI index have no significant improvement. There are still many holes and error parts in them. The results extracted from LJ-PLANUI are closest to the reference data. On the one hand, the extraction results are complete because the missing urban built-up areas are reasonably filled. On the other hand, the error parts of the extraction results are significantly reduced.

In the main urban area of Nanjing, the extraction results from NPP/VIIRS not only have the problem of many false urban built-up areas due to blooming effect and interference light signals but also the problem of lack of urban built-up areas due to lack of local light information (Figure 6a). In Liuhe district and Pukou district, there is boundary expansion due to the blooming effect. At the same time, because some urban built-up areas are not accurately extracted, the extraction results are not continuous and the boundaries are not complete (Figure 6b,c). The problem of discontinuity is particularly serious in the Lishui district and Gaochun district (Figure 6d,e). NPP-VANUI did not improve these problems. Although the extraction results of the Gaochun district extracted from NPP-VANUI increase the patches, the problem of poor continuity has not been solved. The results extracted from NPP-PLANUI are closest to the reference data. The extraction results from NPP-PLANUI in the main urban area are very close to the reference data (Figure 6a) by reducing false extraction and increasing the missing parts. As the missing urban built-up areas increases, the urban built-up areas of the non-main urban area become complete and continuous (Figure 6b–e).

Figure 7 shows the spatial distribution of errors on the extraction results from NTL, VANUI, and PLANUI in the selected regions of Nanjing, China. The urban built-up areas extracted from Luojia 1-01 and LJ-VANUI have many omission pixels inside, and there are many commission pixels around. NPP/VIIRS and NPP/VIIRS generated a large number of omission pixels, which caused the lack of large-scale urban built-up areas, especially in Gaochun and Lishui districts. PLANUI effectively corrects these errors by adding correct pixels and reducing commission pixels.

#### 4.2.2. Accuracy Assessment

Table 2 shows the accuracy assessment of the extraction results from different data. It can be seen that the PLANUI produces the highest extraction accuracy in all study areas among all the data. Specifically, LJ-PLANUI produced 7% higher F1 score than luojia1-01, and NPP-PLANUI produced about 3–4% higher F1 score than NPP/VIIRS in Nanjing and the main urban area. LJ-PLANUI produced 10–14% higher F1 score than luojia1-01 in the selected regions of Liuhe, Pukou, Lishui, and Gaochun. NPP-PLANUI produced about 7–14% higher F1 score than NPP/VIIRS in the selected regions of Liuhe, Pukou, and Lishui. It is worth mentioning that the F1 score of the result extracted from the NPP/VIIRS images is only 0.22 in Gaochun, and NPP-PLANUI increased it to 0.55. Figure 8 shows the kappa value of each extraction result. LJ-PLANUI produced 8–9% higher kappa than luojia1-01 and NPP-PLANUI produced about 5%–6% higher kappa than NPP/VIIRS in Nanjing and the main urban area. The kappa value of PLANUI increased significantly in the non-main urban area of Nanjing. LJ-PLANUI produced 14–17% higher kappa than luojia1-01 in the selected regions of Liuhe, Pukou, Lishui, and Gaochun. NPP-PLANUI produced 10–17% higher kappa than NPP/VIIRS in the selected regions of Liuhe, Pukou, and Lishui. The Kappa of the result extracted from the NPP/VIIRS images is only 0.15 in Gaochun, and NPP-PLANUI increased it to 0.48. This proves that PLANUI is suitable for both NTL with different resolutions and regions with significant differences in development. The accuracy of VANUI is 1–2% generally higher than that of NTL, but significantly lower than that of PLANUI. This shows that PLANUI is better than NTL and VANUI in the urban built-up area extraction.

## 5. Discussion

### 5.1. Advantages of PLANUI

This study proposed PLANUI to extract urban built-up areas. The PLANUI is a new urban index. In it, POI and LST are utilized to characterize the edge and interior of the urban built-up areas and reduce the bloom effect of NTL. PLANUI is the first urban index that combines POI and NTL to extract urban built-up areas. POI is a kind of data from social sensing, while LST is based on natural remote sensing. Moreover, they are all positively correlated with urban built-up areas. With the help of the positive correlation, PLANUI can enhance the signals of urban built-up areas and weaken the influence of non-built-up areas light. The PLANUI was evaluated quantitatively and qualitatively by comparing it with the NTL and the VANUI. The experimental results prove that the PLANUI can not only effectively reduce the blooming effect, but also improve the extraction accuracy, which achieved the first goal of this study. Comparison with NTL data and VANUI showed that the PLANUI not only increases the boundary and detail information of the area affected by the blooming effect but also adds many missing urban built-up areas. The extraction results from the PLANUI demonstrate great integrity and connectivity and are consistent with the reference built-up areas (Figure 6), which is a big advantage of the PLANUI. In addition, the PLANUI gained significant effects in districts where economic development varies. In the main urban area, it can guarantee high extraction accuracy, while supplementing detailed information and modifying boundary information. In the non-main urban area, it not only significantly improved the extraction accuracy but also complemented the missing parts. These prove that the PLANUI has extensive applicability in the field of urban built-up area extraction, which is another advantage of the PLANUI.

### 5.2. Difference for the Indexes Based on Luojia-1 and NPP/VIIRS

The applicability of PLANUI to different NTL data is great. By contrast, LJ-PLANUI performs better than NPP-PLANUI, in terms of the spatial distribution of built-up areas and extraction accuracy. The experimental results of Luojia 1-01 and NPP/VIIRS showed that they have different shortcomings. The results extracted from the NPP/VIIRS images have a significant blooming effect. They are excessively concentrated in the main urban area, has insufficient information on the boundary and details, and have many missing parts in the administrative area with a low level of development. Although the extraction results from the Luojia 1-01 images can well reflect the entire urban built-up areas of Nanjing, there remains a problem with many holes and misinformation. The PLANUI based on NTL images at two different resolutions achieved good results. The LJ-PLANUI can fill excessive holes and add missing parts, and increase connectivity and integrity of the urban built-up areas. The NPP-PLANUI can solve problems such as the out-of-bounds boundary caused by the blooming effect by supplementing the boundary information, which guarantees the complexity of urban built-up areas. Therefore, PLANUI is suitable for both NTL with different resolutions and regions with significant differences in development. Due to the higher spatial resolution of Luojia 1-01 than NPP/VIIRS, LJ-PLANUI performs better than NPP-PLANUI in the details of the interior and boundary of the urban built-up areas (Figure 5). The urban built-up areas extracted from LJ-PLANUI are more complete and accurate (Figure 6). It can be seen from Table 2 that the accuracy of the results extracted based on the two indices is relatively high and the difference between the two is generally small. Only in Gaochun District, the extraction accuracy of LJ-PLANUI is significantly higher than that of NPP-PLANUI, which shows that LJ-PLANUI is more suitable for areas with low economic development than NPP-PLANUI.

### 5.3. Applications of PLANUI

As the urbanization develops rapidly, the challenge becomes how to make a reasonable policy for urban planning, to ensure the orderly expansion of the city to avoid rapid population growth, environmental pollution and resource shortage [41,42,43,44]. Accurate data of urban built-up areas are necessary for urban planning [36,45], so it is very important to extract urban built-up areas effectively. The extraction results from PLANUI has advantages of high precision and consistency with the reference built-up area. In addition, PLANUI is suitable for the extraction in the regions with different levels of development, greatly improving extraction efficiency and accuracy. Therefore, PLANUI can be applied to urban planning and has certain guiding significance for urban expansion, urban structure, and urban governance. The positive correlation between POI, LST, and urban built-up areas, which PLANUI is based on, is more stable than the negative correlation of vegetation index. As a social and economic complex, the city has two basic characteristics, namely the agglomeration effect and scale effect, which can be reflected by the dynamic density of POI [18]. LST can reflect the phenomenon that the temperature in urban built-up areas is generally higher than that of vegetation, water bodies, or other land types [22,23]. It has a positive correlation with the distribution of urban built-up areas. The stability of the relationship between vegetation index and the urban built-up areas is affected by the following two points. First, in a short period of time, the urban vegetation cover changes in a large area. Most cities in China are in the stage of transformation, development, and construction. In a short period of time, large-scale land development and land use transformation are very common, which has a great impact on urban vegetation coverage. Second, not all regions urbanize through large-scale vegetation cover reduction. In suburban and desert cities in North America, such as Las Vegas and Nevada, vegetation for residential and commercial would increase in the process of urbanization [16]. Therefore, in theory, PLANUI is suitable for more cities than the index based on vegetation, which will be verified in the next study. From DMSP/LOS to NPP/VIIRS to Luojia 1-01, the development of NTL is characterized by the improvement of spatial resolution. In this trend, PLANUI is suitable for NTL with different resolutions, which provides a theoretical basis to introduce more NTL with high spatial resolution into PLANUI. It can be seen that PLANUI has a broad application space in the field of urban built-up area extraction from NTL in the future.

### 5.4. Uncertainties and Prospects

The PLANUI is intuitive and easy to implement. POI is a kind of important geospatial big data, which has advantages over remote sensing and population density data in terms of update speed and acquisition cost [18]. LST is introduced into the study as traditional remote sensing data. The method of average value is utilized to establish the PLANUI in this paper. The method utilizes the advantages of the three and obtained good results. The study of the next stage needs to explore the deeper physical relationship of the three and choose the more appropriate method to combine the data to obtain more accurate results of urban built-up areas. However, PLANUI also has some shortcomings. The establishment of the PLANUI is based on the positive correlation between NTL, POI, LST data, and urban built-up areas. The positive correlation is quite significant in the most urban built-up areas, which ensures high accuracy of the results extracted from the PLANUI. But in some regions, there is uncertainty about the positive correlation, which is the leading cause of the error. For example, in the areas surrounding the extremely developed urban built-up areas, the positive correlation could be influenced and strengthened by the surrounding environment. In urban built-up areas with extremely slow development, the positive correlation could be relatively weak. In future research, more data sources are considered to be introduced, especially the data, such as NDVI and EVI, with a significant negative correlation with urban built-up areas. The kind of negative correlation might correct the uncertainty of the error producing area and improves accuracy. Future research tries to combine more diverse data with NTL data to find a method with broader applicability and higher precision. The PM2.5 and road network data are considered for future researches. In recent years, people have paid more and more attention to air pollution and governance. As an essential indicator for monitoring air pollution, PM2.5 has become a reliable data source for related research. Due to the influence of many pollution sources and poor air circulation, the urban built-up areas usually have higher PM2.5 concentrations than the surrounding areas [46]. Road network density is an essential criterion for measuring the level of urbanization [47]. Due to different economic development levels, the urban built-up areas have higher road network density than the surrounding areas. These laws provide a reliable theoretical basis for combining PM2.5 data and road network data with NTL images to extract urban built-up areas.

## 6. Conclusions

This study proposed the PLANUI, a new index combining NTL, POI, and LST, to overcome the limitations of the blooming effect and low resolution of NTL data and to characterize urban built-up areas to improve the accuracy of the extraction.

(1)Compared with the VANUI index, the PLANUI can make the extraction results closer to the reference data in overall shape and detail information, and can significantly improve the accuracy of the urban built-up areas extracted.(2)The PLANUI has extensive applicability, both for regions with varying degrees of economic development and NTL data with different resolutions. In the main urban area, PLANUI can increase the boundary and internal details while ensuring high accuracy. In the non-main urban area, PLANUI can increase the extraction accuracy by adding missing urban built-up areas. LJ-PLANUI can fill the holes inside the urban built-up areas and reduce falsely extracted parts. NPP-PLANUI can significantly reduce the overflow effect to solve the problem of border expansion and can also make up the lack of urban built-up area information.(3)Due to the fact that Luojia 1-01 images have a higher spatial resolution than the NPP/VIIRS images, LJ-PLANUI is better than NPP-PLANUI in showing the details of the interior and boundary of the urban built-up areas and performs better than NPP-PLANUI in the areas with poor economic development.(4)LJ-PLANUI has achieved more significant accuracy improvement than NPP-PLANUI, which shows that PLANUI is suitable for high-resolution NTL data. In the future, PLANUI can be utilized with more high-resolution night light data to conduct built-up area extraction research, so it has a broad application prospect. Moreover, PLANUI can provide an effective approach for research on urban built-up area extraction and contribute to the research investigating urban expansion, urban planning, and urban pattern governance.

## Figures and Tables

**Figure 1 sensors-20-02918-f001:**
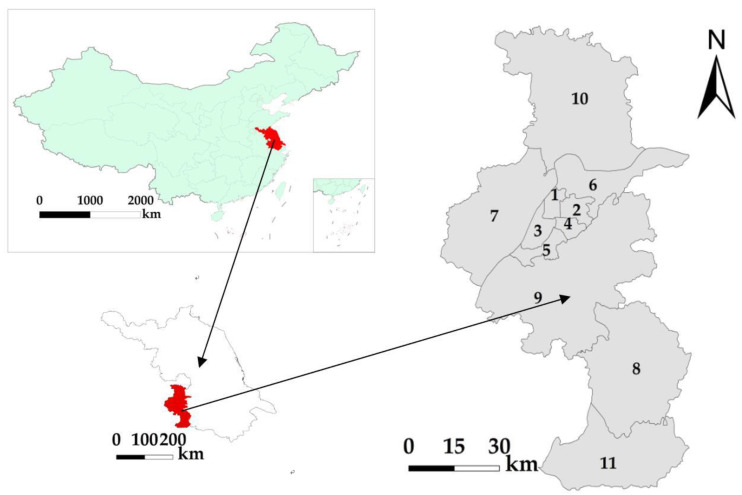
Administrative divisions in Nanjing: (**1**) Gulou; (**2**) Xuanwu; (**3**) Jianye; (**4**) Qinhuai; (**5**) Yuhuatai; (**6**) Qixia; (**7**) Pukou; (**8**) Lishui; (**9**) Jiangning; (**10**) Liuhe; (**11**) Gaochun.

**Figure 2 sensors-20-02918-f002:**
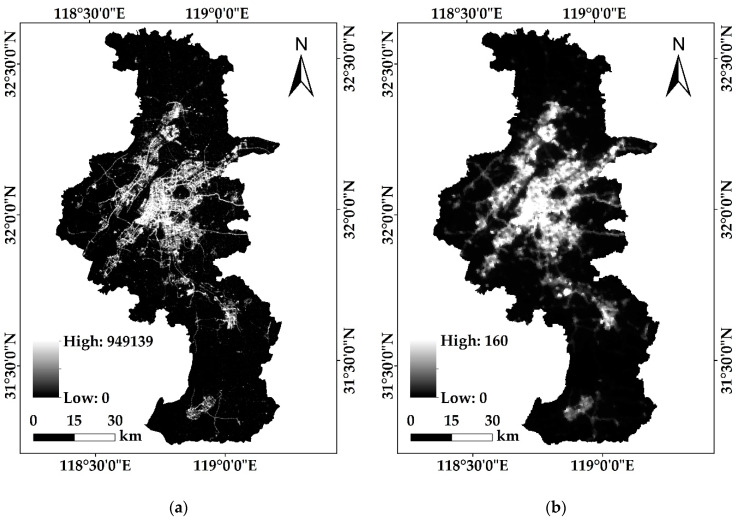
Original data of study area: (**a**) Luojia 1-01 image; (**b**) NPP/VIIRS image; (**c**) LST data; (**d**) Kernel Density Estimation result of POI.

**Figure 3 sensors-20-02918-f003:**
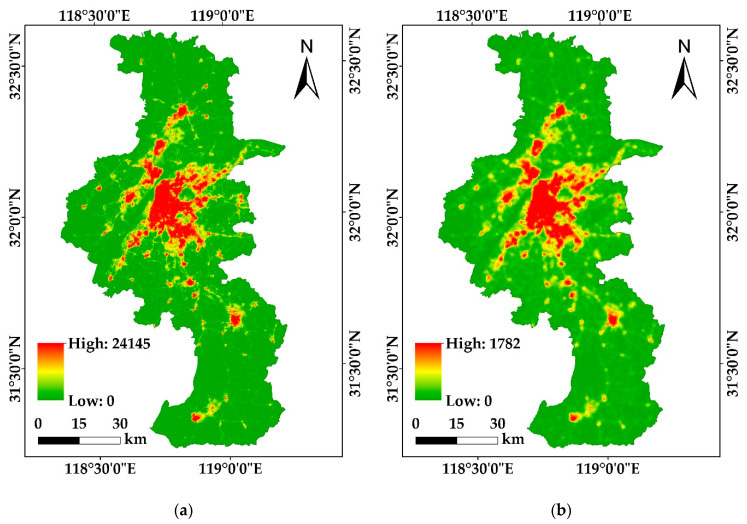
PLANUI images of the study area: (**a**) LJ-PLANUI; (**b**) NPP-PLANUI.

**Figure 4 sensors-20-02918-f004:**
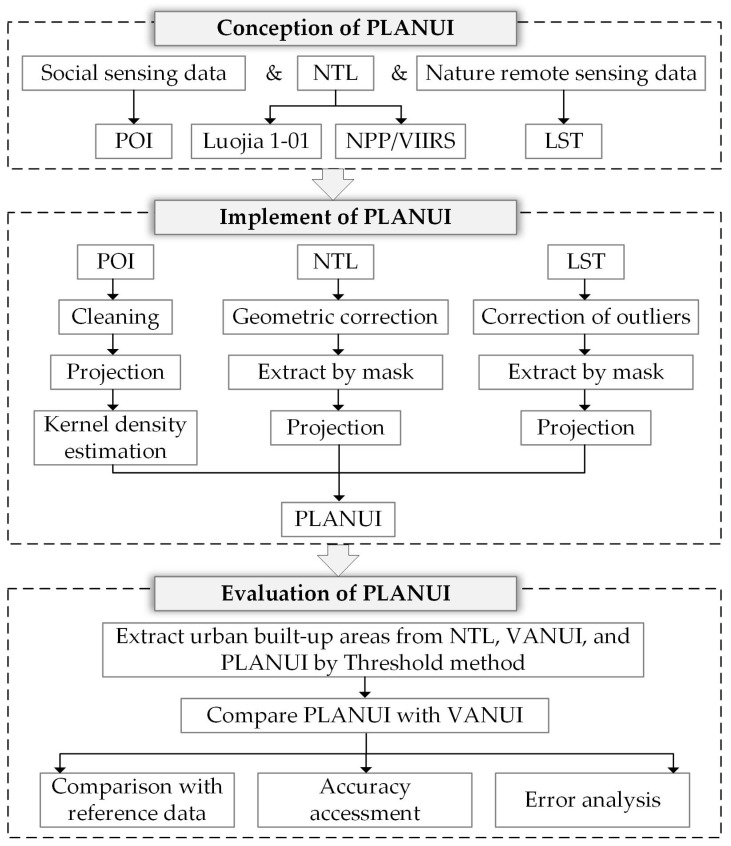
The workflow of conception, implement and evaluation of PLANUI for urban built-up area extraction.

**Figure 5 sensors-20-02918-f005:**
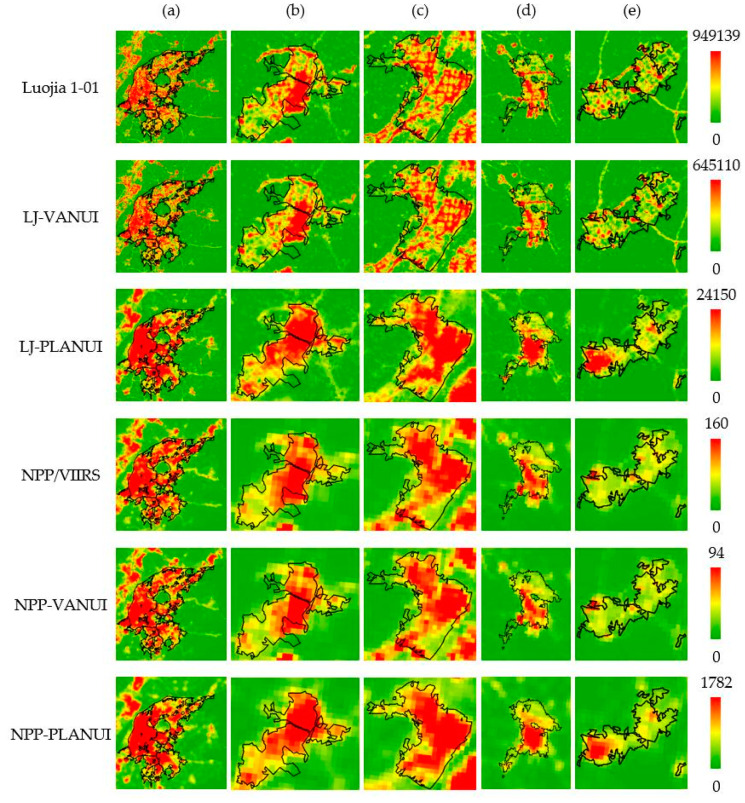
The comparison among NTL, VANUI, and PLANUI images in the selected regions in Nanjing by reference data (black lines): (**a**) Main Urban Area; (**b**) Liuhe; (**c**) Pukou; (**d**) Lishui; (**e**) Gaochun.

**Figure 6 sensors-20-02918-f006:**
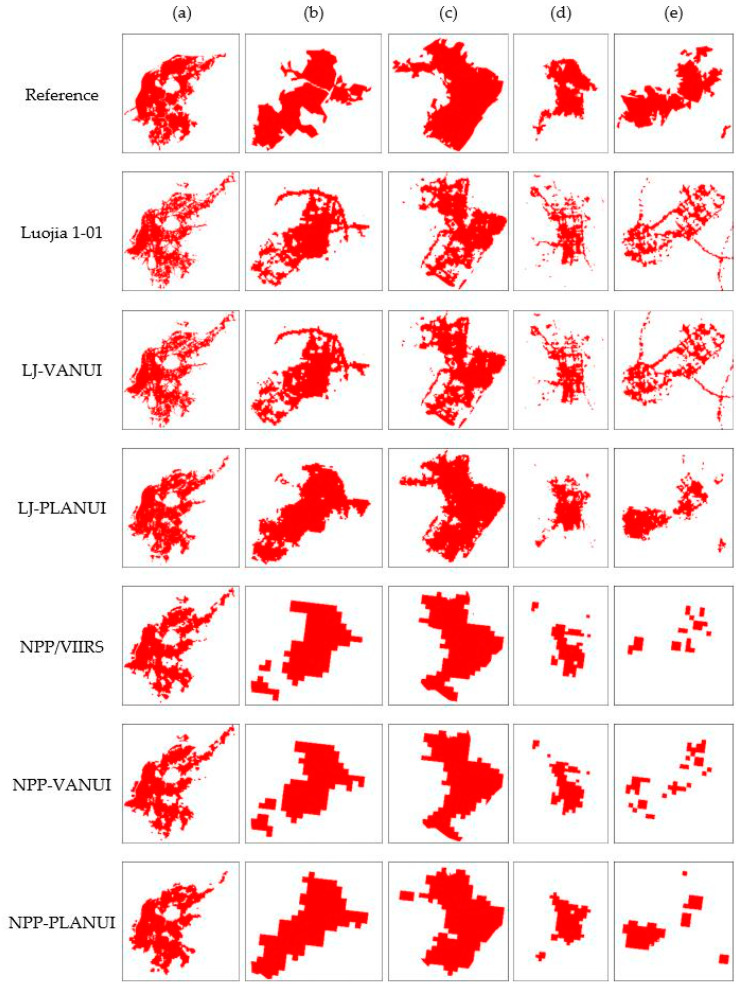
Comparison between reference data and extraction results from NTL, VANUI, and PLANUI in the selected regions in Nanjing: (**a**) Main Urban Area; (**b**) Liuhe; (**c**) Pukou; (**d**) Lishui; (**e**) Gaochun.

**Figure 7 sensors-20-02918-f007:**
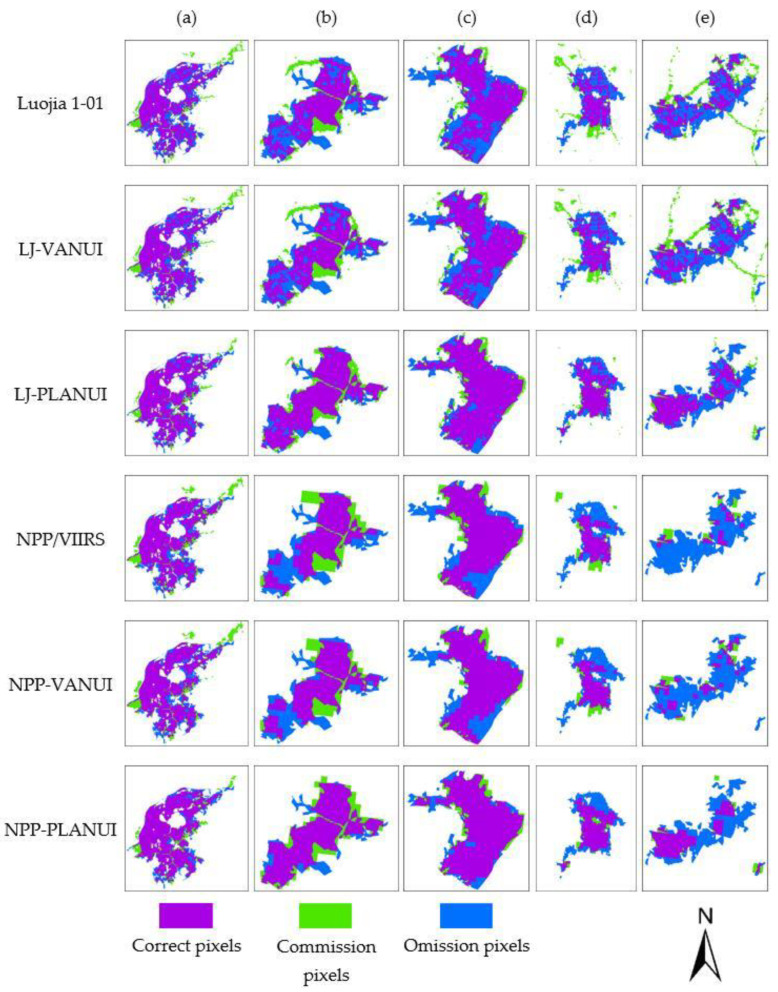
Error analysis for the extraction results from NTL, VANUI, and PLANUI in the selected regions in Nanjing: (**a**) Main Urban Area; (**b**) Liuhe; (**c**) Pukou; (**d**) Lishui; (**e**) Gaochun.

**Figure 8 sensors-20-02918-f008:**
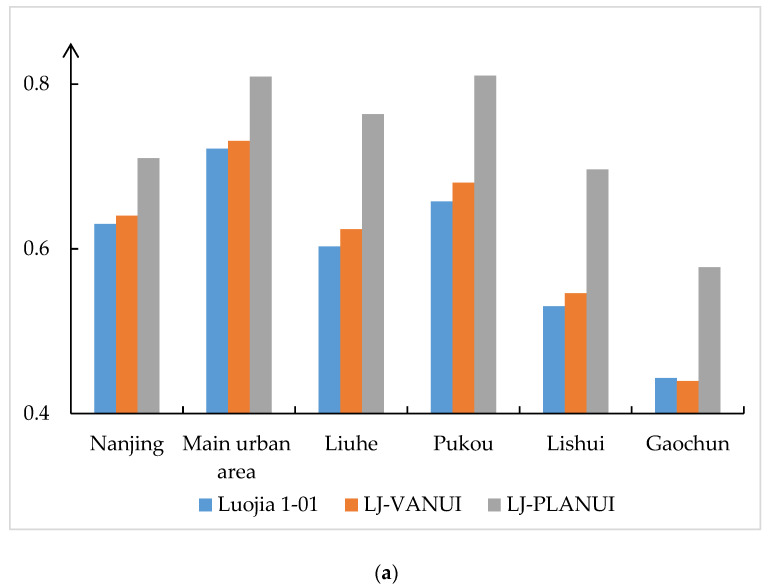
Comparison of Kappa coefficients of extraction results based on (**a**) Luojia 1-01 and (**b**) NPP/VIIRS in Nanjing and in the five selected regions: Main Urban Area, Liuhe, Pukou, Lishui, and Gaochun.

**Table 1 sensors-20-02918-t001:** Introduction to Specifications of Luojia 1-01 and NPP-VIIRS Images.

Satellite	Luojia 1-01	NPP-VIIRS
Spatial Resolution	130 m	750 m
Width	250 km	3060 km
Spectrum Range	0.46–0.98 μm	0.5–0.9 μm
Radiometric Resolution	14 bits	14 bits
Available Years	June 2018–present	November 2011–present

**Table 2 sensors-20-02918-t002:** Accuracy assessment on the extraction results.

Study Area	Index	Luojia 1-01	LJ- VANUI	LJ- PLANUI	NPP/VIIRS	NPP- VANUI	NPP- PLANUI
	Precision	0.73	0.74	0.81	0.78	0.79	0.81
Nanjing	Recall	0.63	0.64	0.70	0.67	0.68	0.71
	F1-score	0.68	0.69	0.75	0.72	0.73	0.76
Main	Precision	0.84	0.84	0.88	0.82	0.82	0.87
urban	Recall	0.77	0.78	0.86	0.85	0.85	0.88
area	F1-score	0.80	0.81	0.87	0.84	0.84	0.87
	Precision	0.79	0.79	0.82	0.74	0.78	0.80
Liuhe	Recall	0.63	0.65	0.83	0.65	0.68	0.86
	F1-score	0.70	0.71	0.83	0.69	0.73	0.83
	Precision	0.89	0.90	0.91	0.89	0.92	0.91
Pukou	Recall	0.67	0.69	0.85	0.76	0.76	0.87
	F1-score	0.76	0.78	0.88	0.82	0.83	0.89
	Precision	0.69	0.71	0.89	0.76	0.79	0.89
Lishui	Recall	0.52	0.53	0.63	0.46	0.48	0.59
	F1-score	0.60	0.61	0.74	0.58	0.60	0.71
	Precision	0.66	0.64	0.88	0.62	0.73	0.88
Gaochun	Recall	0.46	0.47	0.51	0.13	0.23	0.39
	F1-score	0.54	0.54	0.64	0.22	0.34	0.55

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
