# Peer review of "A POI and LST Adjusted NTL Urban Index for Urban Built-Up Area Extraction"

_sensors, 2020, doi:10.3390/s20102918_

Round 1

Reviewer 1 Report

This study proposes the PLANUI index from NTL images, POI, and LST data to extract urban areas. And the results were tested and verified by other data. The study is well organized and easy to follow, as well as the results are within the scope of the field of sensors. However, I have a question that only Nanjing was chosen as the experimental case, and the reliability of the results is questionable. Can the PLANUI extracted results be extended to other cities? More cities should be selected for urban area extraction and verification. Also, the reference format is incorrect. For example, Zhang [14] should be replaced by Zhang et al. [14]. Please consider the following literatures.

Evaluating the ability of NPP-VIIRS nighttime light data to estimate the gross domestic product and the electric power consumption of China at multiple scales: A comparison with DMSP-OLS data

Evaluation of NPP-VIIRS nighttime light composite data for extracting built-up urban areas

Modeling spatiotemporal CO2 (carbon dioxide) emission dynamics in China from DMSP-OLS nighttime stable light data using panel data analysis

Author Response

Dear reviewer: Firstly, we appreciate you for your comments and suggestions. We all think the Comments are professional and they are really useful for us to receive and improve the article. According to your comments, we revised the article. The revised explanations are as follows: Comments and suggestions from reviewer 1: This study proposes the PLANUI index from NTL images, POI, and LST data to extract urban areas. And the results were tested and verified by other data. The study is well organized and easy to follow, as well as the results are within the scope of the field of sensors. Point 1: However, I have a question that only Nanjing was chosen as the experimental case, and the reliability of the results is questionable. Can the PLANUI extracted results be extended to other cities? More cities should be selected for urban area extraction and verification. Response 1: The reasons for choosing Nanjing as the research area in this article are mentioned on lines 117-123. Nanjing has a total area of 6,587 square kilometers and the characteristics of a large north-south span. The Gaochun District in the south is 140 kilometers away from Liuhe District in the north. Nanjing consists of 11 administrative regions, which are at different levels of development. In other words, Nanjing can be regarded as a collection of many cities. This paper has selected five regions with different levels of economic development to verify the applicability of PLANUI. The results show that: PLANUI has achieved good results in both the main urban area with high economic development level and the non-main urban area with poor economic development level, which indicates that it has good applicability and can be used in other cities. Point 2: Also, the reference format is incorrect. For example, Zhang [14] should be replaced by Zhang et al. [14]. Please consider the following literatures… Response 2: All similar wrong reference formats have been thoroughly checked and corrected (lines 48, 58, 61, 66, 95, 97, 202, 227). We have carefully read the articles you recommend, and have carefully borrowed the research experience in the process of article revision. These articles have been added to the reference. Thanks again for your comments valuable Suggestions, I hope that you’re satisfied with the revisions.

Reviewer 2 Report

It is a clearly written paper with a transparent structure. The article intends to produce the ready-to-use index to identify urban built-up areas. I would recommend it for publication in the journal after the following comments will have been addressed.

All abbreviations, even if they were explained in the Abstract, should be explained, i.e. the full name should be used, on their first mention the text of the paper.

The authors recommend using Land Surface Temperature as a component of the proposed index based on the Urban Heat Island (UHI) effect. However, the UHI effect does not work in all the cases. How the authors deal with such phenomena as urban cool island/urban cool lake, which may form in different conditions (see, for example, https://doi.org/10.1002/joc.4747 and https://doi.org/10.1016/j.uclim.2015.09.001) and with heatwaves, which level the temperature inside and outside cities (see, for instance, https://doi.org/10.1007/s00704-008-0088-3).

It is clear that the denser the urban area, the higher the accuracy assessment coefficients. I would ask the authors to provide, in a table, the values of the accuracy assessment indexes for each of the five urban areas separately, as well as to discuss in more detail the differences between the areas. Also, confusion matrices should be included in the accuracy assessment for some images to better illustrate the accuracy assessment process.

Author Response

Dear reviewer:

Firstly, we appreciate you for your comments and suggestions. We all think the Comments are professional and they are really useful for us to receive and improve the article. According to your comments, we revised the article. The revised explanations are as follows:

Comments and suggestions from reviewer 2: It is a clearly written paper with a transparent structure. The article intends to produce the ready-to-use index to identify urban built-up areas. I would recommend it for publication in the journal after the following comments will have been addressed.

Point 1: All abbreviations, even if they were explained in the Abstract, should be explained, i.e. the full name should be used, on their first mention the text of the paper.

Response 1: After checking the full article, all abbreviations have been explained by the full name of them on their first mention in the text of the paper (lines 34, 40, 41, 54, 55, 59…).

Point 2: The authors recommend using Land Surface Temperature as a component of the proposed index based on the Urban Heat Island (UHI) effect. However, the UHI effect does not work in all the cases. How the authors deal with such phenomena as urban cool island/urban cool lake, which may form in different conditions (see, for example, https://doi.org/10.1002/joc.4747 and https://doi.org/10.1016/j.uclim.2015.09.001) and with heatwaves, which level the temperature inside and outside cities (see, for instance, https://doi.org/10.1007/s00704-008-0088-3).

Response 2: LST is data that can reflect the spatial distribution of land surface temperature. It is the result of the combined action of natural and human effects, such as urban heat island effect, urban cold island effect, urban cold lake effect, and heatwave effect. LST has a positive correlation with the distribution of urban built-up areas [22,23], so many studies [14,24,25] used LST data to assist in the extraction of urban built-up areas and achieved good results. It proves that LST has a good ability to extract urban built-up areas, so this paper utilized LST to assist in extracting built-up areas too. Utilizing LST is based on the positive correlation between LST and the distribution of urban built-up areas. I am sorry that the previous expression was inaccurate and caused a misunderstanding.

Point 3: It is clear that the denser the urban area, the higher the accuracy assessment coefficients. I would ask the authors to provide, in a table, the values of the accuracy assessment indexes for each of the five urban areas separately, as well as to discuss in more detail the differences between the areas. Also, confusion matrices should be included in the accuracy assessment for some images to better illustrate the accuracy assessment process.

Response 3: The values of the accuracy assessment indexes for each of the five urban areas are provided in Table 2 (line 387). More details of the differences between the areas are also discussed on lines 312-319. The Kappa Coefficients of five urban areas calculated from the confusion matrix are shown in Figure 8. The discussion about Kappa is on lines 320-328.

Thanks again for your comments valuable Suggestions, I hope that you’re satisfied with the revisions.

Reviewer 3 Report

The study aim is to present the research on “A POI and LST adjusted NTL urban index for urban built-up area extraction.” The paper is well written, and I like to propose a few comments and minor revisions of this paper. 

  1. Introduction well written.
  2. The limitation of the study needs to highlighted because you have used several data with several spatial resolutions.
  3. How does this approach can be used in other study areas?
  4. The conclusion needs to revise.

Author Response

Dear reviewer:

Firstly, we appreciate you for your comments and suggestions. We all think the Comments are professional and they are really useful for us to receive and improve the article. According to your comments, we revised the article. The revised explanations are as follows:

Comments and suggestions from reviewer 3: The study aim is to present the research on “A POI and LST adjusted NTL urban index for urban built-up area extraction.” The paper is well written, and I like to propose a few comments and minor revisions of this paper. 

Point 1: Introduction well written.

Response 1: We have checked the introduction, revised some of the content, polished the language, and corrected the spelling errors (lines 34-114).

Point 2: The limitation of the study needs to highlighted because you have used several data with several spatial resolutions.

Response 2: Among NTL, POI, and LST data, LST has a relatively low spatial resolution. LST is data that can reflect the spatial distribution of surface temperature. It can be seen from Figure 2c that the spatial heterogeneity of LST is not strong and the regional change is not large. Increasing the spatial resolution of LST does not significantly improve the results. Therefore, this article did not highlight the limitations of the study.

Point 3: How does this approach can be used in other study areas?

Response 3: The reason for choosing Nanjing as the research area is that it is representative (mentioned on lines 117-123). Nanjing has a total area of 6,587 square kilometers and the characteristics of a large north-south span. The Gaochun District in the south is 140 kilometers away from Liuhe District in the north. Nanjing consists of 11 administrative regions, which are at different levels of development. In other words, Nanjing can be regarded as a collection of many cities. This paper has selected five regions with different levels of economic development to verify the applicability of PLANUI. The results show that: PLANUI has achieved good results in both the main urban area with high economic development level and the non-main urban area with poor economic development level, which indicates that it has good applicability and can be used in other study areas.

Point 4: The conclusion needs to revise.

Response 4: The conclusion is divided into 4 parts to elaborated separately. Some new content was added to the conclusion (lines 455-469), such as the evaluation of PLANUI and the difference for the index based on Luojia-1 and NPP / VIIRS.

Thanks again for your comments valuable Suggestions, I hope that you’re satisfied with the revisions.

Round 2

Reviewer 1 Report

I don't have any more comments for this revision.

Reviewer 2 Report

Dear authors,

Thank you for the answers provided! I believe that the improvements made to the article significantly improved the quality of the descriptions. I would recommend it for publication in the journal in the current form.

Best,